# Medical imaging utilization in migrants compared with nonmigrants in a universal healthcare system: A population-based matched cohort study

**Giancarlo Di Giuseppe**[1], **Rinku Sutradhar**[1,2], **Priscila Pequeno**[2], **Marilyn L. Kwan**[3], **Diana L. Miglioretti**[4,5], **Rebecca Smith-Bindman**[6], **Jason D. Pole**[1,2,7] *

**1** Dalla Lana School of Public Health, University of Toronto, Toronto, Ontario, Canada, **2** ICES, Toronto, Ontario, Canada, **3** Division of Research, Kaiser Permanente Northern California, Pleasanton, California, United States of America, **4** Division of Biostatistics, Department of Public Health Sciences, University of California, Davis, California, United States of America, **5** Kaiser Permanente Washington Health Research Institute, Seattle, Washington State, United States of America, **6** Department of Epidemiology and Biostatistics, University of California San Francisco, San Francisco California, United States of America, **7** Centre for Health Sciences Research, University of Queensland, Brisbane, Australia

* j.pole@utoronto.ca

**Data Availability Statement:** The dataset from this study is held securely in coded form at ICES. While

## Abstract

### Background

Medical imaging is an integral part of healthcare. Globalization has resulted in increased mobilization of migrants to new host nations. The association between migration status and utilization of medical imaging is unknown.

### Methods and findings

A retrospective population-based matched cohort study was conducted in Ontario, Canada from April 1, 1995 to December 31, 2016. A total of 1,848,222 migrants were matched 1:1 to nonmigrants in the year of migration on age, sex, and geography. Utilization of computed tomography (CT), magnetic resonance imaging (MRI), radiography, and ultrasonography was determined. Rate differences per 1,000 person-years comparing migrants to nonmigrants were calculated. Relative rates were calculated using a recurrent event framework, adjusting for age, sex, and time-varying socioeconomic status, comorbidity score, and access to a primary care provider. Estimates were stratified by migration age: children and adolescents (≤19 years), young adults (20 to 39), adults (40 to 59), and older adults (≥60). Utilization rates of CT, MRI, and radiography were lower for migrants across all age groups compared with Ontario nonmigrants. Increasing age at migration was associated with larger differences in utilization rates. Older adult migrants had the largest gap in imaging utilization. The longer the time since migration, the larger the gap in medical imaging use. In multivariable analysis, the relative rate of imaging was approximately 20% to 30% lower for migrants: ranging from 0.77 to 0.88 for CT and 0.72 to 0.80 for MRI imaging across age groups. Radiography relative rates ranged from 0.84 to 0.90. All migrant age groups, except older adults,

legal data sharing agreements between ICES and data providers (e.g., healthcare organizations and government) prohibit ICES from making the dataset publicly available, access may be granted to those who meet pre-specified criteria for confidential access by contacting Data & Analytic Services, available at www.ices.on.ca/DAS (email: das@ices.on.ca). The full dataset creation plan and underlying analytic code used to generate results cannot be made publicly available due to legal concerns. Those who meet pre-specified criteria may be granted access to the code by contacting ICES Data & Analytic Services, understanding that the computer programs may rely upon coding templates or macros that are unique to ICES and are therefore either inaccessible or may require modification.

**Funding:** This study was supported by the National Cancer Institute at the National Institutes of Health (R01CA185687 to RSB, MK and DM and R50CA211115 to RSB, MK and DM). In addition, this was supported by ICES, which is funded by an annual grant from the Ontario Ministry of Health, Long-Term Care. The study is based on data and/or information compiled and provided by the Canadian Institute for Health Information. The funders had no role in the design and conduct of the study; collection, management, analysis, and interpretation of the data; preparation, review, or approval of the manuscript; and decision to submit the manuscript for publication.

**Competing interests:** The authors have declared that no competing interests exist.

**Abbreviations:** ADG, Aggregated Diagnosis Groups; CI, confidence interval; CT, computed tomography; MRI, magnetic resonance imaging; OHIP, Ontario Health Insurance Plan; RR, relative rate; SD, standard deviation.

had higher rates of ultrasonography. The indication for imaging was not captured, thus it was not possible to determine if the imaging was necessary.

## Conclusions

Migrants utilized less CT, MRI, and radiography but more ultrasonography. Older adult migrants used the least amount of imaging compared with nonmigrants. Future research should evaluate whether lower utilization is due to barriers in healthcare access or health-seeking behaviors within a universal healthcare system.

---

### Author summary

#### Why was this study done?

- Globalization has led to an increasing number of migrants who must navigate a new healthcare system in the countries they migrate to.

- Migrants have been shown to use specific health services, such as cancer screening or general practitioner use, at a lower rate than those born in the host country.

- Limited studies have been performed which have evaluated the longitudinal usage patterns of medical imaging of migrants compared with nonmigrants.

#### What did the researchers do and find?

- A retrospective cohort study of 1,848,222 migrants to Ontario, Canada were matched to an equal number of nonmigrants and evaluated for the utilization rates of computerized tomography, magnetic resonance imaging, radiography, and ultrasonography medical imaging.

- Usage rates of medical imaging were 20% to 30% lower in migrants compared with nonmigrants.

- A longer time since migration resulted in larger disparities in imaging use, with the largest age-related differences in utilization patterns observed in older migrants.

#### What do these findings mean?

- Large disparities in the utilization of medical imaging within this universal healthcare system exist which warrants further investigation to uncover mechanisms of action.

- These data may be used by healthcare policymakers and practitioners in regions where migration plays a significant role in its population.

## Introduction

Globalization has led to increased migration to developed nations over the last several decades. Despite this resettlement, discernible variations in health system utilization patterns exist

between migrant and nonmigrant populations, often with migrants using fewer medical services [1]. A propensity of host nations to implement standards regarding the health condition of prospective migrants has also resulted in the phenomenon known as the "healthy immigrant effect," whereby immigrants tend to exhibit favorable general health [2,3]. Empirical evidence originating from several countries with differing healthcare systems have demonstrated migrants have lower utilization of health services, including mental health resources and cancer screening [4–8]. While previous research has primarily focused on examining specific types of health services utilized by migrants, a paucity of evidence remains evaluating the longitudinal utilization of general radiologic imaging within a universal healthcare system.

Frameworks for healthcare use and health-seeking behaviors in migrants have been well described [9,10]. Health systems and governmental policies provide macro-structural factors for migrants, while the delivery style of care in the host jurisdiction, such as volume and costs, contribute to utilization [9]. Migrant-specific factors, such as the degree of assimilation or cultural beliefs about health, influence a migrant's propensity to seek care and perceptions of the need for services [10]. Furthermore, the length of time a migrant has resided in the host country can positively influence their health-seeking behaviors, as longer duration is often associated with increased familiarity with the healthcare system and greater social integration [9]. Understanding the use of medical imaging among migrant groups is crucial for preventing downstream consequences such as late disease detection or poor health outcomes [8,11,12].

Migrants are faced with several challenges when navigating a new healthcare system. Language, cultural, socioeconomic, and system factors are often cited by migrants as reasons for barriers to healthcare access [13–15]. Migrants must also acquire a general practitioner in their new host nation, and many report difficulties finding one, often resulting in the underutilization of primary care [16–18]. These dynamics can have significant consequences on the use of medical imaging. Recent literature has illustrated that disparities in radiology use exist for migrant and ethnic groups [19,20], highlighting an important area of research into health service use patterns. Thus, it is crucial to gain insight into any disparities in migrant populations for policy-making and healthcare planning and to ensure equity in the delivery of medical imaging [21–23].

Evaluating the utilization of radiologic services of migrants is well suited to be conducted in Canada. Access to necessary medical care is readily available to all citizens and residents independent of insurance coverage or financial payments for service. Conducting such evaluations in Canada circumvents the financial burdens and systemic challenges often encountered in other countries, providing a more efficient and equitable assessment environment. In the community setting, patients must first consult with a general practitioner to obtain a referral for medical imaging. Given the socialized healthcare system, the nation possesses population-based databases capturing nearly all interactions of individuals receiving care using the publicly funded system. Canada experiences high rates of annual migration and is one of the most ethnically diverse countries in the world [24]. Recent estimates from the 2021 Canadian Census report that 30% of Ontario, Canada's population are migrants [25]. Despite migrants comprising a large proportion of the province's population, a knowledge gap exists evaluating the use of medical imaging in migrants compared with nonmigrants.

This study aimed to evaluate the health services utilization of medical imaging in migrants and examined the use of computerized tomography (CT), magnetic resonance imaging (MRI), radiography, and ultrasonography spanning 2 decades for migrants within a universal healthcare system. Our objective was to determine the association between migration status and medical imaging utilization, and the extent to which this association differs by the age at which individuals migrate and the length of time since migration. We hypothesized that a longer residency in the nation would be associated with similar utilization rates between migrants and nonmigrants.

## Methods

This study adhered to the Strengthening the Reporting of Observational Studies in Epidemiology (STROBE) reporting guideline (S1 Checklist) [26]. Ethics approval was obtained from the University of Toronto, Ontario, Canada with a waiver of consent (Protocol Number 33470).

### Population and study design

A population-based matched cohort study was performed using linked administrative databases. Data on CT, MRI, radiography, and ultrasonography utilization from 01 April 1995 to 31 December 2016 were obtained for individuals eligible for the Ontario Health Insurance Plan (OHIP), a governmentally funded universal healthcare plan available for all Canadian citizens, permanent residents, and newly landed migrants in Ontario (available after a 3-month waiting period). Individuals of all ages migrating and landing in Ontario, Canada between 01 April 1995 and 31 December 2016 were identified. Migrants were assigned an index date that was the earlier of their landing date or start of OHIP eligibility. In Ontario, migrants may apply for OHIP while their permanent residency application is being processed by the federal government, and thus the earlier of these 2 dates would ensure capture of interactions within the healthcare system.

In the year of their migration (observation start date), migrants were individually matched to nonmigrants of Ontario one-to-one without replacement on age, sex, and geography. The forward sortation area was used as the measure of geographical matching and is the first 3 digits of the Canadian postal code. This geographic measure is used by the federal postal service and is proportional to population density. Nonmigrants were assigned the same index date as their matched migrant pair. Individuals were followed from the index date until they were censored on the loss of OHIP eligibility (moved out of the province), death, or 31 December 2016 whichever occurred first.

### Data sources

Migrants were identified from the Immigration, Refugees and Citizenship Canada database, which is managed by the federal government of Canada and contains all Canadian migration records. Immigration data were available from 1985 onwards. The Immigration, Refugees and Citizenship Canada database has been shown to have high linkage rates to population-based healthcare data [27]. Information captured by the database includes personal identifying information, age at time of landing, sex, and landing date, as well as sociodemographic information including country of origin, marital status, education, and migrant class. Migrants were classified as economic class (e.g., business class or entrepreneurs, skilled workers), family class (e.g., familial sponsorship), resettled refugees and protected persons, or other unclassifiable migrants.

A control cohort of nonmigrants was identified from the Registered Persons Database, which contains basic sociodemographic and geographical residence data for anyone who has ever been eligible for the governmentally funded universal healthcare plan. Nearly all of Ontario's 14-million population, including migrants, is contained in the Registered Persons Database, which is managed by the Ministry of Health and Long-term Care. Upon relocation out of the province, individuals are no longer eligible for health insurance and must notify the Ministry. An individual was considered a nonmigrant if they were a Canadian-born citizen or migrated to Canada prior to 1985, as no Immigration, Refugees, and Citizenship Canada data was available prior to this date.

Medical imaging data were captured through linkages to fee-for-service physician billing records to the Ministry of Health from the OHIP database, inpatient hospitalization imaging

from the Canadian Institute for Health Information's Discharge Abstract Database, and emergency department imaging from the National Ambulatory Care Reporting System. This encompasses all medical imaging performed within the province, except for imaging provided by private practices, which are uncommon. These data sets were linked using unique encoded identifiers and analyzed at ICES.

## Imaging utilization and covariates

Utilization for CT, MRI, radiography, and ultrasonography modalities were evaluated for all examination types. No restriction was imposed on where the imaging occurred (i.e., hospital, clinic). Imaging was captured using billing codes supplied by the OHIP Schedule of Benefits and Fees or the Canadian Classification of Health Interventions [28,29]. To prevent overcounting of duplicate billing codes between the linked data sets, only a single imaging event per modality per day was included. We did not include radiation treatment planning, image processing, or imaging reinterpretation. Detailed methods of these deduplication methods have been described elsewhere [30].

Comorbidity burden was described using the Johns Hopkins Adjusted Clinical Group system derived from diagnostic codes associated with physician billing records, emergency department visits, or hospitalizations. For each participant, the number of Aggregated Diagnosis Groups (ADG), which describes 32 diagnosis clusters of overall morbidity burden [31], was calculated using data for the 2 years prior to index date. Since migrants did not have 2 years of look-back healthcare data to assign an ADG score at their index landing date, their first calculable ADG score after 2 years in the province was used to backfill the initial study period. This approach was performed to adjust for ADG in multivariable regression analyses in the first 2 years of observation and to avoid starting our observation window 2 years after landing. We were interested in immediate utilization after landing and assumed that within the first 2 years in the province, migrant comorbidity would not change meaningfully [32]. This assumption was confirmed by calculating the average comorbidity score for migrants and nonmigrants during follow-up which showed no major comorbidity changes in the nonmigrant group within the first 2 years of observed follow-up (S1 Fig).

Access and rostering to a primary care provider were collected using a 2-year look back window to determine the presence or absence of being rostered to a general practitioner. This served as a proxy to account for access to healthcare, as this may confound imaging rates given primary care physicians act as gatekeepers to the healthcare system. Socioeconomic status was estimated by linking postal codes from the Registered Persons Database to the Canadian Census using the Postal Code Conversion File provided by Statistics Canada to assign neighborhood urban income quintiles and rural status [33].

## Statistical analysis

Descriptive statistics were calculated using means and standard deviations (SDs) for continuous variables and frequencies and proportions for categorical data. Differences in the distribution of baseline characteristics at index were compared between migrants and the matched cohort using standardized differences, with a value >0.10 considered imbalanced [34]. Age at migration was categorized as: children and adolescents (0 to 19 years), young adults (20 to 39 years), adults (40 to 59 years), and older adults ($\geq$60 years). Incidence rates were calculated as the number of imaging events per 1,000 person-years of observation. Rate differences and 95% confidence intervals (95% CI) for each imaging modality outcome were calculated for migrants relative to nonmigrants and adjusted for baseline age, sex, index year, and geographical region due to prior matching. To examine if length of time since migration influenced

imaging use, incidence rates and rate differences were stratified by time periods from the index date as: 0 to <5, 5 to <10, 10 to <15, 15 to <20, and ≥20 years from migration.

A priori, medical imaging use was deemed to vary throughout the life-course and health statuses, which we believed would result in time intervals between imaging occurrences to be potentially correlated within individuals. Thus, a recurrent event framework was used to account for any potential clustering or correlation among multiple exams within the same individual. A mean cumulative function curve was estimated to illustrate the mean cumulative number of imaging exams for each modality since index date. An Anderson–Gill recurrent event multivariable regression model was implemented, stratified by age at migration, to determine the association between migrant status and each imaging modality [35,36]. Unlike the Poisson regression which assumes event rates are constant over time, the Anderson–Gill framework considers the timing of repeated events to be non-homogenous and models event rates as a function of time.

Regression analyses were adjusted for age, sex, socioeconomic status, comorbidity burden ADG score, and visit to a primary care provider. Covariates with missing data had a separate missing category added in the regression analyses. Socioeconomic status, ADG score, and visits to a primary care provider 2 years prior were updated annually and treated as time-varying covariates in the regression model. Relative rates (RR) and 95% CIs were reported for migrants compared with the matched cohort. All analyses were stratified by age group to examine age-related effects.

Age- and sex-stratified multivariable regression models were performed *ad hoc* to explore the effect of sex on the relative rate of imaging. Additionally, an ad hoc analysis was performed by stratifying the regression models by age at index and the grouped year of migration (1995 to 1999, 2000 to 2004, 2005 to 2009, and 2010 to 2016). These time-stratified models contained an interaction between migration status and sex. All analyses were performed in SAS, version 9.4 at ICES, Toronto, Ontario, Canada.

### Sensitivity analyses

Several sensitivity analyses were performed to ensure robustness of the observed results. First, all age-stratified multivariable regression analyses were re-run by shifting the index date forward by 2 years for migrants and the matched cohort to ensure an ADG score and access to a primary care provider could be calculated for migrants using 2 years of follow-up data following their arrival in Canada. Second, restriction of analyses to necessary imaging defined as head CT in participants with a traumatic brain injury seen in the emergency department. This analysis was undertaken given we believed there should be little difference in medical imaging use for this indication based on migration status. Emergency department visits for a traumatic brain injury were identified using validated ICD codes [37,38] and head CT was assessed within 7-days of the injury. The proportion and number of CT exams per injury were calculated.

### Results

A total of 2,165,142 migrants were identified during the study period; 316,920 were excluded (14.6%) because an ADG score could not be determined due to a lack of interaction with the healthcare system, leaving 1,848,222 migrants eligible for study inclusion and matched to an equal number of nonmigrants. The average age at baseline was 29.9 (SD, 17.1) years, with 47.1% of participants migrating as young adults and 27.9% as children or adolescents. Just over half of the cohort was female. Differences were observed between migrants and nonmigrants in socioeconomic position at baseline, with a larger proportion of migrants belonging to the lowest income quintile (Table 1). Migrants of the economic class represented 50.8% of

**Table 1. Population characteristics of migrants and matched Ontario, Canada nonmigrants at baseline migration date.**

| Characteristic | Migrant | | Nonmigrant | | Standardized difference |
|---|---|---|---|---|---|
| | *N* | (%) | *N* | (%) | |
| **Total cases (*N*)** | 1,848,222 | | 1,848,222 | | |
| **Age (years)** | | | | | |
| Mean ± SD | 29.9 | (17.1) | 29.9 | (17.1) | 0.00 |
| 0–19 | 515,874 | (27.9%) | 515,654 | (27.9%) | 0.00 |
| 20–39 | 869,977 | (47.1%) | 870,252 | (47.1%) | 0.00 |
| 40–59 | 339,781 | (18.4%) | 339,882 | (18.4%) | 0.00 |
| ≥60 | 122,590 | (6.6%) | 122,434 | (6.6%) | 0.00 |
| **Sex** | | | | | |
| Female | 971,472 | (52.6%) | 971,472 | (52.6%) | 0.00 |
| Male | 876,750 | (47.4%) | 876,750 | (47.4%) | 0.00 |
| **Socioeconomic status** | | | | | |
| Urban Income Quintile 1 (lowest) | 697,923 | (37.8%) | 475,312 | (25.7%) | 0.26 |
| Urban Income Quintile 2 | 419,586 | (22.7%) | 419,878 | (22.7%) | 0.00 |
| Urban Income Quintile 3 | 307,315 | (16.6%) | 356,444 | (19.3%) | 0.07 |
| Urban Income Quintile 4 | 234,501 | (12.7%) | 301,347 | (16.3%) | 0.10 |
| Urban Income Quintile 5 (highest) | 167,452 | (9.1%) | 271,324 | (14.7%) | 0.17 |
| Rural | 17,497 | (1.0%) | 18,647 | (1.0%) | 0.01 |
| *missing* | 3,948 | (0.2%) | 5,270 | (0.3%) | 0.01 |
| **Migrant category** | | | | | |
| Economic class | 938,319 | (50.8%) | – | – | |
| Family class | 612,955 | (33.2%) | – | – | |
| Resettled refugee | 262,324 | (14.2%) | – | – | |
| Other | 34,609 | (1.9%) | – | – | |
| *missing* | 15 | (0.0%) | 1,848,222 | (100.0%) | |
| **Johns Hopkins Aggregated Diagnosis Groups** | | | | | |
| Mean ± SD | 3.8 | (2.8) | 4.2 | (3.1) | 0.13 |
| **Primary care provider visit in 2 years prior to baseline** | | | | | |
| No | 1,562,438 | (84.5%) | 287,714 | (15.6%) | 1.91 |
| Yes | 285,784 | (15.5%) | 1,560,508 | (84.4%) | 1.91 |
| **Index year** | | | | | |
| 1995–1999 | 397,558 | (21.5%) | 397,558 | (21.5%) | 0.00 |
| 2000–2004 | 515,095 | (27.9%) | 515,095 | (27.9%) | 0.00 |
| 2005–2009 | 456,842 | (24.7%) | 456,842 | (24.7%) | 0.00 |
| 2010–2016 | 478,727 | (25.9%) | 478,727 | (25.9%) | 0.00 |
| **Follow-up time (years)** | | | | | |
| Mean ± SD | 11.2 | (5.9) | 10.6 | (6.0) | 0.10 |
| <10 years | 802,359 | (43.4%) | 864,682 | (46.8%) | 0.07 |
| 10–20 years | 907,929 | (49.1%) | 865,997 | (46.9%) | 0.05 |
| ≥20 years | 137,934 | (7.5%) | 117,543 | (6.4%) | 0.04 |
| **Reason for censoring** | | | | | |
| Death | 27,361 | (1.5%) | 65,739 | (3.6%) | 0.13 |
| Loss of provincial Healthcare Plan Eligibility | 83,185 | (4.5%) | 157,326 | (8.5%) | 0.16 |
| End of study | 1,737,676 | (94.0%) | 1,625,157 | (87.9%) | 0.21 |

–, not applicable; SD, standard deviation.

the cohort, followed by 33.2% in the family class and 14.2% as resettled refugees. Migrants had a significantly lower ADG mean score of 3.8 (SD, 2.8) compared with 4.2 (SD, 3.1) in nonmigrants and had fewer visits to primary care providers. Migrants and nonmigrants were followed for an average of 11.2 (SD, 5.9) and 10.6 (SD, 6.0) years, respectively. Overall, 27,361 (1.5%) migrants died during follow-up, whereas death occurred in 65,739 (3.6%) nonmigrants.

Migrants had lower mean cumulative number of imaging exams for CT, MRI, and radiography across all age groups, with the difference in rates widening over time (Fig 1). Cumulative imaging rates were higher with increasing patient age and age at migration and increased over time for all age groups and modalities. Differences were smallest among the youngest individuals. Only young adult migrants received greater cumulative imaging for ultrasonography. Mean cumulative number of imaging events in 5-year intervals for each modality stratified by age at migration is provided in S1 Table, showing a widening in the difference in imaging occurrences.

### Overall incidence rates of utilization

Overall utilization rate differences per 1,000 person-years varied across age groups (Fig 2 and S2 Table). Utilization rates for CT, MRI, and radiographic imaging were lower for all migrant age groups and as the age at migration increased, the rate difference increased. Older adult migrants had the largest difference in rates for all modalities and the largest reduction in utilization was observed in older adult migrants for radiography, who had a 530.0 (95% CI: 527.1, 532.9) per 1,000 person-years fewer exams. Young adult migrants had higher rates of ultrasonography (rate difference, 55.5; 95% CI: 54.9, 56.1).

### Imaging utilization for time since migration

Imaging utilization by age at migration, and stratified by time since migration, are presented in Fig 3 and S3 Table. With increasing patient age utilization rates increased, but as the time since migration increased, the magnitude of the relative absolute reduction in imaging relative to nonmigrants also grew larger (top half of graphs). For example, among older adults aged ≥60 years when they were new migrants (0 to <5 years since migration) the rate difference for CT imaging was −106.2 (95% CI: −107.9, −104.5) exams per 1,000 person-years and increased to −282.8 (95% CI: −308.6, −256.9) when ≥20 years since migration (Fig 3A). For younger adults, a similar widening difference in rates with time since migration was observed for MRI utilization (Fig 3B).

Migrants utilized less radiography for all time periods, with the rate difference increasing in magnitude for all age groups except for children and adolescents as time since migration increased (Fig 3C). In children and adolescent migrants, the rate difference slightly decreased from −59.6 (95% CI: −60.5, −58.8) to −48.4 (95% CI: −56.2, −40.7) radiography exams per 1,000 person years from the 0 to <5 to ≥ 20-year time period.

Utilization of ultrasound use was similar for all age groups and all times since migration, other than for the oldest aged adults. Young adult migrants even utilized ultrasonography at a slightly higher rate. In older adults, the magnitude of the rate difference increased as time elapsed, resembling that seen for the other imaging modalities (Fig 3D).

### Multivariable recurrent event model of imaging

The RR of CT, MRI, and radiography imaging was significantly lower among migrants in all age groups in the multivariable analyses (Table 2). For CTs, the RR of imaging among migrants ranged from 0.77 (95% CI: 0.77, 0.78) to 0.80 (95% CI: 0.79, 0.80). Similar lower RR of MRIs was seen across all age groups with older adults using this modality the least (RR, 0.72;

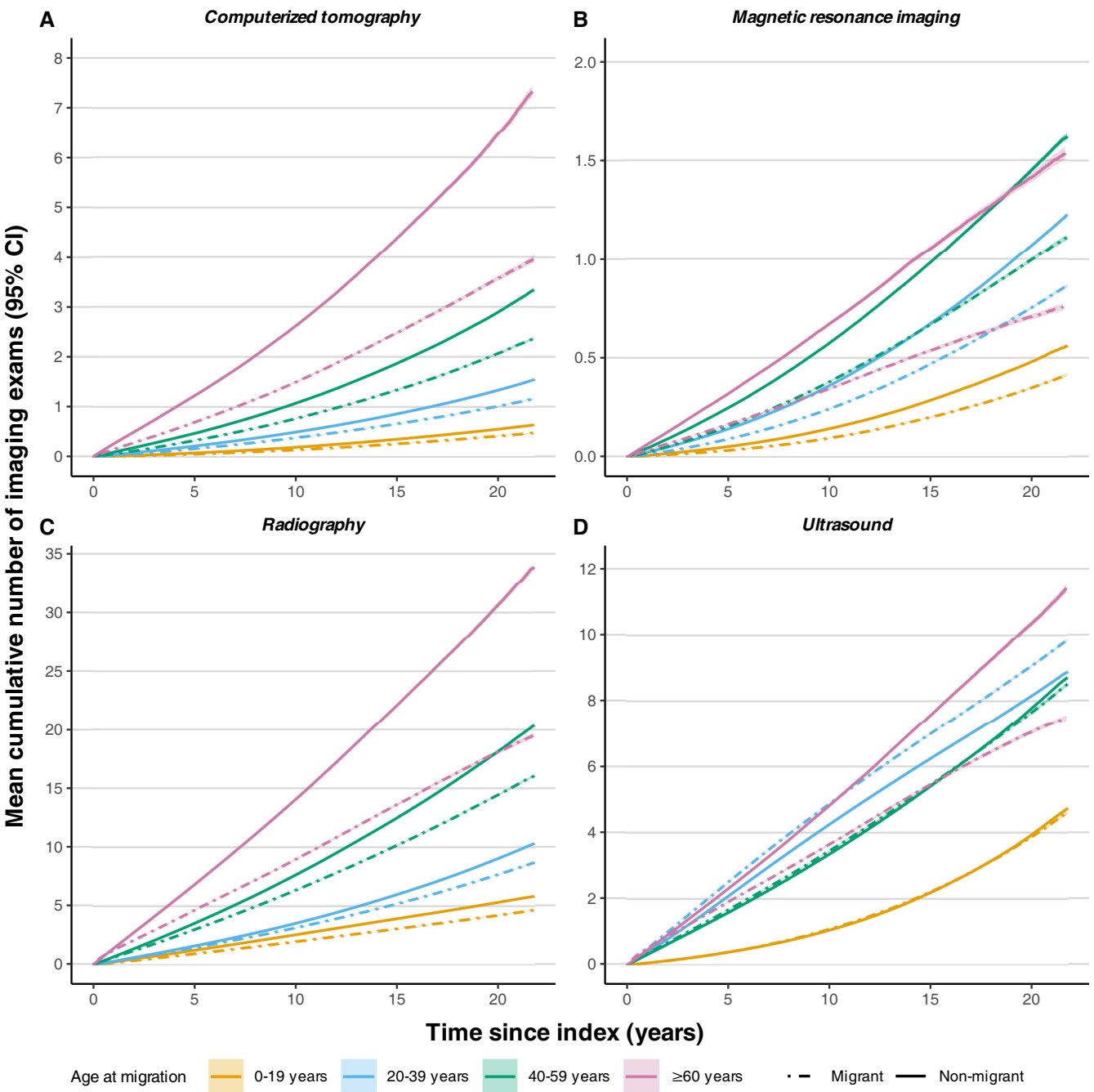

**Fig 1. Mean cumulative number of recurrent imaging events stratified by age at migration with accompanying 95% CIs.** Adjusted for baseline matching variables: age, sex, index year, and geographical area. Index is the migration date for the migrant cohort and the dummy migration date for matched individuals.

95% CI: 0.71, 0.73). The RR for radiography was similar, with a more modest decrease, ranging from 0.90 (95% CI: 0.90, 0.90) for adults to 0.84 (95% CI: 0.84, 0.84) for older adults.

Migrant adults, young adults, and children had higher RRs of ultrasonography, which ranged from 1.08 (95% CI: 1.08, 1.09) to 1.21 (95% CI: 1.21, 1.21). Older adult migrants had a slightly lower RR of ultrasonography (RR, 0.98; 95% CI: 0.98, 0.98).

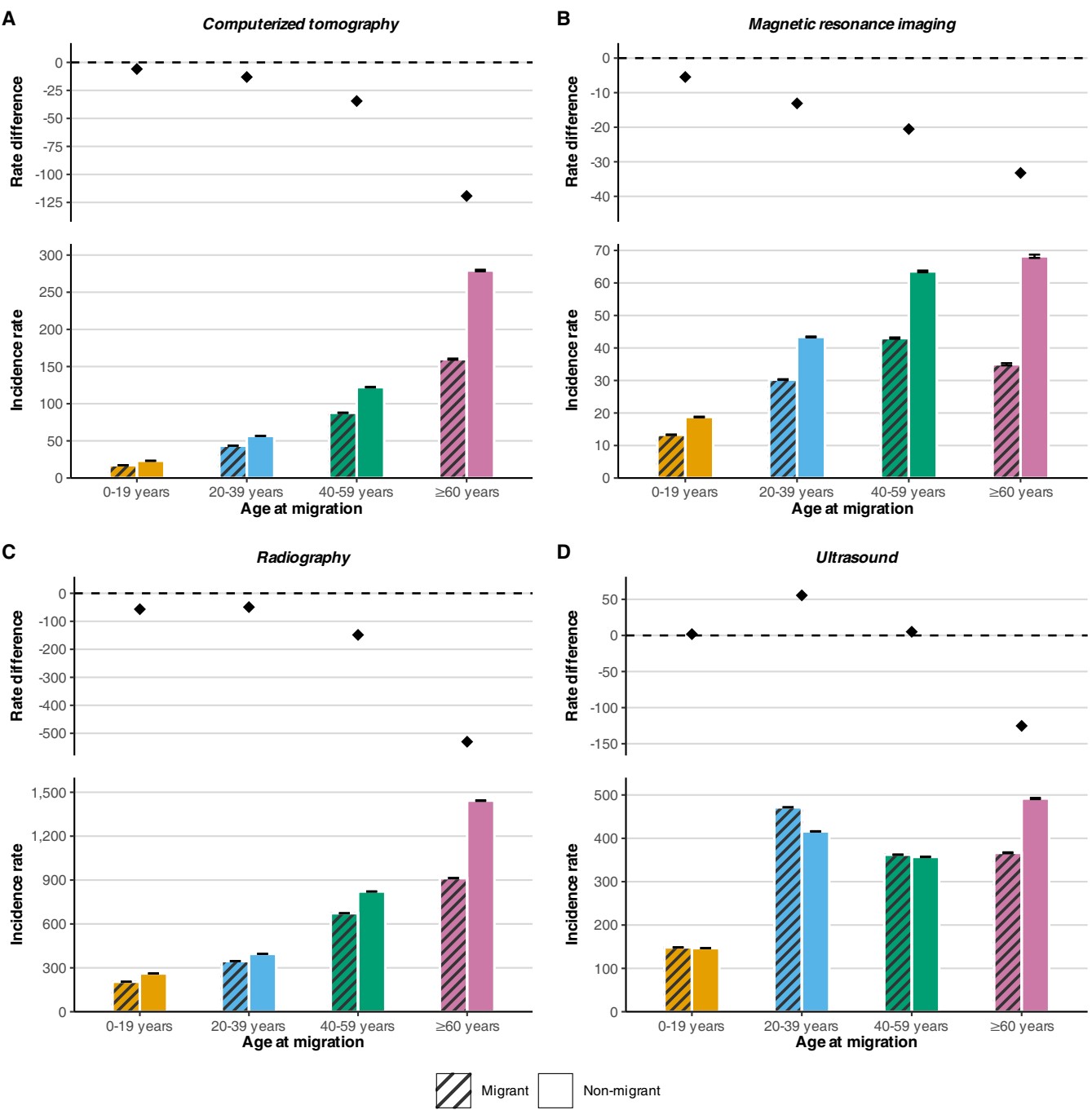

**Fig 2. Overall incidence rate and absolute rate difference of medical imaging utilization for migrants compared with nonmigrants with accompanying 95% CIs.** Incidence rate and rate difference values per 1,000 person-years. Rate difference was calculated as the absolute difference between migrants and nonmigrants. Dashed lined represents a null effect of the rate difference. Adjusted for baseline matching variables: age, sex, index year, and geographical area.

The age and sex stratified models revealed utilization of less medical imaging was similar for males and females across most age groupings and imaging modalities (S4 Table). Only a few percentage points differed between the relative rates of males and females. The largest sex differences were observed in older aged migrants for CT imaging with males using less than females. Aside from both males and females having higher use of ultrasonography over secular

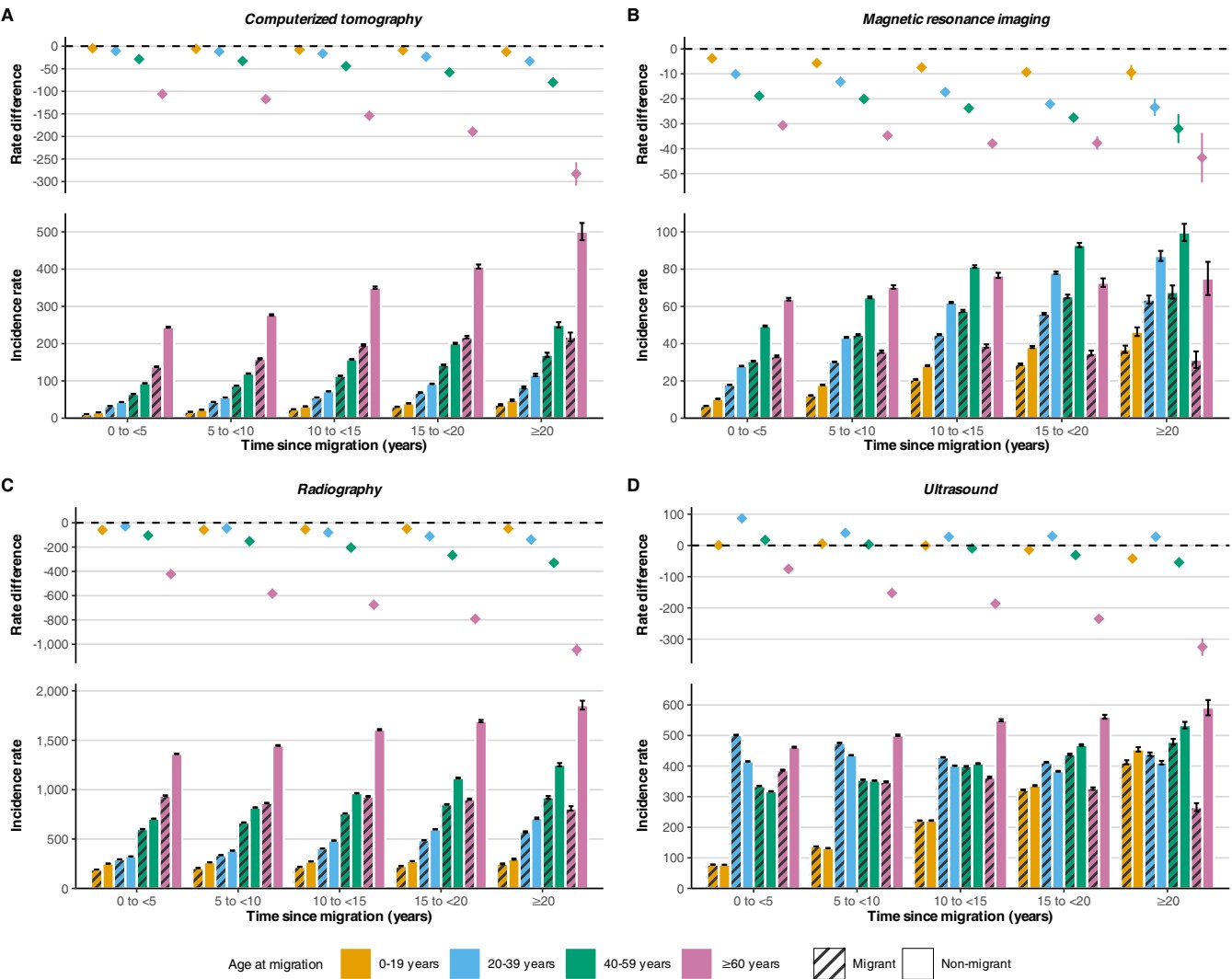

**Fig 3. Incidence rate and absolute rate difference of medical imaging utilization by elapsed time since migration with accompanying 95% CIs.** Incidence rate and rate difference values per 1,000 person-years. Rate difference was calculated as the absolute difference between migrants and nonmigrants. Dashed lined represents a null effect of the rate difference. Adjusted for baseline matching variables: age, sex, index year, and geographical area.

**Table 2. Relative rate of medical imaging utilization stratified by age at migration.**

| Age at migration | Relative rate (95% CI) | | | |
|---|---|---|---|---|
| | Computerized tomography | Magnetic resonance imaging | Radiography | Ultrasound |
| 0–19 years | 0.80 (0.79 to 0.80) | 0.80 (0.79 to 0.81) | 0.86 (0.86 to 0.86) | 1.08 (1.08 to 1.09) |
| 20–39 years | 0.77 (0.77 to 0.78) | 0.74 (0.74 to 0.74) | 0.89 (0.89 to 0.89) | 1.21 (1.21 to 1.21) |
| 40–59 years | 0.79 (0.79 to 0.80) | 0.79 (0.78 to 0.79) | 0.90 (0.90 to 0.90) | 1.13 (1.12 to 1.13) |
| ≥60 years | 0.78 (0.77 to 0.78) | 0.72 (0.71 to 0.73) | 0.84 (0.84 to 0.84) | 0.98 (0.98 to 0.98) |

Age-stratified models represent the relative rate of imaging for migrants compared with the reference group of matched nonmigrants. Models are adjusted for age, sex, index year, and time-varying socioeconomic status, Aggregated Diagnostic Group score, and visits to a primary care provider. Time-varying covariates were updated annually until the end of observation.

95% CI, 95% confidence interval.

time, no temporal effect was observed when considering the relative rates by migration period (S5 Table).

In sensitivity analyses shifting the index date 2 years forward for both migrants and the matched cohort, the multivariable regression results were substantively unchanged and showed a slightly larger RR for most imaging modalities except for ultrasound use (S6 Table). When focusing only on indicated imaging (head CTs for a traumatic head injury), there were no differences in the mean number of CTs between migrants and nonmigrants (S2 Fig).

## Discussion

Migrants utilize less medical imaging than nonmigrants within a universal healthcare system. Use of CT, MRI, and radiography is approximately 20% to 30% lower among migrants. While this finding was observed across all age groups, the magnitude of the absolute reduction in utilization was greatest among the oldest subjects, and in the longest time since migration. The only exception was for ultrasound where migrants have the same or even higher use of ultrasound. Patterns of utilization differ by age at migration and time since migration and likely reflect a combination of differing healthcare needs and health-seeking behaviors.

An unexpected finding was that the length of time since migration resulted in a larger difference in imaging rates between migrants and nonmigrants, rather than a dissipation of differences over time. Our sex and migration period stratified analyses further illustrate no major sex differences in use, and that the disparities seen in utilization of medical imaging have not improved over time. Insights provided by this study support the importance of addressing disparities in healthcare utilization of migrants, which may be instructive for policy-makers and practitioners [23,39,40]. Thus, it is essential to place our findings within Levesque and colleagues multidimensional framework of healthcare access which encompasses approachability, acceptability, availability/accommodation, affordability, and appropriateness [41]. Each of these dimensions plays a critical role in shaping how populations interact with the healthcare system.

Imaging utilization results observed in this study align with previous population-based studies in migrants which have illustrated lower utilization of other medical services in Canada [42–44], the US [45–47], and Europe [48,49]. Landed migrants must navigate a new healthcare system and common themes of language barriers, lack of information, and cultural differences may act to impede interactions with the healthcare system and lead to lower healthcare services utilization [15]. Migrants also report a barrier to healthcare access given the need to adjust to a new healthcare system [50,51]. These barriers may hinder migrants' perceptions of healthcare acceptability given cultural/social factors or the approachability of the new host nation's healthcare system in their ability to seek care.

To account for the barriers to access in our analysis, we adjusted for socioeconomic status and visits to a primary care provider, the gatekeepers to the medical system, yet differences persist. Migrants continued to exhibit lower imaging rates after considering these proxies as barriers to access. This suggests a need for future research to explore other mechanisms that may aid in explaining the lowered utilization of medical imaging or if there is heterogeneity in the effects of being a migrant on utilization across factors are effect modifiers such as having a PCP or income. Within Canada's universal healthcare system, migrants are not required to have insurance coverage or pay for medical imaging. However, indirect costs such as traveling to healthcare centres may contribute to the affordability of migrants seeking medical imaging. Migrants may also return to their country of origin to seek medical care [52]. However, this is unlikely within our study since we controlled for primary care access and may be more prominent in other jurisdictions with alternate healthcare systems.

Further, we hypothesized that the longer a migrant lived within a jurisdiction with a universal healthcare system, the more likely their medical imaging utilization would become comparable to nonmigrants of the nation because of general assimilation. Surprisingly, the contrary was observed with a widening difference in the utilization rates as time since migration elapsed and migrants used even less imaging. Migrants who have lived longer in their new jurisdiction are more likely to report having a primary care provider compared with recent migrants [53]. However, this did not result in their having equal use of imaging compared with nonmigrants. This suggests that passively waiting for access for new migrants to improve over time does not occur and highlights an opportunity for interventions targeted at bolstering health literacy among migrants [54].

Strong age-related effects were observed in imaging utilization. Specifically, older aged adults at the time of migration had the largest dissimilarity to rates seen in nonmigrants, whereas differences were much smaller in younger migrants. Generally, older patients use more medical imaging as their health status declines with aging. However, even after adjusting for access to primary care or comorbidity status, substantial differences were observed for older migrants. Older migrants face unique challenges to access to primary healthcare [55], which exacerbate their medical imaging use. Indeed, adult older adult migrants may have already established lower health-seeking behaviors, which may aid in explaining the sustained lowered medical imaging use [56].

Ultrasonography was the only modality which was equal or higher in migrants. Detailed data on indication were not available; however, we observed that the largest increase in ultrasound use occurred in young adults of reproductive age. Similarly, the highest utilization rates were within the earliest time period from the arrival date. Thus, this increased use may be reproductive medical imaging during the family formative years. Recent literature within a universal healthcare system has illustrated that female migrant populations utilize infertility consult services more than the general population [57].

Our sensitivity analysis aimed to explore the possible variation in CT utilization rates in patients with suspected traumatic brain injury according to their migration status. This indication entails a low imaging threshold owing to its high yield of important clinical findings that have a direct impact on patient outcomes. There was no reduction in imaging utilization by migration status for this essential indication for imaging. Patients with traumatic brain injury are seen in the emergency department where staff decide to image, and these results suggest the decision for imaging for this acute injury does not consider migration status or related factors associated with the differences seen in imaging rates overall. Other imaging described in this report may be more discretionary, where both patient health-seeking behavior and physician discretion each play a role, and where migration status is an important factor. While structural racism, unconscious bias, and inherent patient preferences for imaging may all play a role, further investigation is required to confirm their contribution to this disparity.

This study has several strengths, including its large size, universal access to healthcare, and complete population capture of interactions within the healthcare system. Long-term follow-up was available to allow assessment of patterns of care over time with respect to years since migration. The regression models adjusted for comorbidity score as a proxy for overall health of participants in our study, thereby adjusting for the healthy immigrant effect and the observation that migrants tend to be healthier than nonmigrant populations [58]. A recurrent event analytical approach was used to account for the dynamic nature of health status without assuming a constant imaging event rate over time and considered the fluidity of imaging for healthcare needs. Socioeconomic status, comorbidity score, and access to a primary care provider were also treated as time-varying covariates in our regression models to account for any changes in access or health throughout follow-up. We also performed several sensitivity analyses and found the observed results were robust except for imaging for suspected traumatic brain injury.

The study has several limitations. Imaging utilization was restricted to one imaging exam per modality per day to prevent double counting and overestimating utilization across the multiple sources of information, thus we believe our estimates are conservative but represent a clinically meaningful outcome of unique days of imaging utilization. We were unable to account for imaging prescribed in outpatient settings but not ultimately performed. The indication for imaging was not captured, thus it was not possible to determine if the imaging was necessary. Future research on imaging utilization in migrant groups should consider detailed clinical indications for imaging to elucidate the specific reasons driving the observed utilization rates. Family composition was not captured, and it may be possible that children of older migrants or other familial members can aid in any language barriers experienced when navigating the healthcare system and influence imaging use. The Immigration, Refugees and Citizenship Canada database began in 1985, and migrants landing before that date might have been included in the matched cohort. If a migrant landing before 1985 was included as a matched population comparator, this would likely have led to an underestimation of the gap in imaging of migrants. Lastly, we did not consider race or ethnic origin, and thus could not determine if these factors influenced utilization rates. Previous research from the US has illustrated that diagnostic imaging utilization differs by patient race and ethnicity [59].

Migrants have a lower utilization of CT, MRI, and radiographic medical imaging. A longer length of time since migration resulted in a larger negative difference in utilization rates. This observation highlights an important discrepancy in imaging utilization in a universal health system where disparities in healthcare are unexpected. Addressing disparities in migrant medical imaging use requires a multidimensional lens incorporating both healthcare system factors and patient-level perspectives regarding access and beliefs so that strategies can ensure equitable access to healthcare for migrant populations. Evidence from this study provides valuable guidance to healthcare policymakers in regions where migration plays a significant role in shaping the composition of both current and future populations. Future research should explore the reasons for this disparity and whether they impact patient health outcomes, particularly within a universal healthcare system.

## Supporting information

**S1 Checklist. STROBE Statement—Checklist of items that should be included in reports of cohort studies.**
(PDF)

**S1 Table. Mean cumulative number of imaging exams every 5 years of follow-up stratified by age at migration.**
(PDF)

**S2 Table. Number of images, rates, and measures of effect for imaging utilization of migrants and nonmigrants.**
(PDF)

**S3 Table. Imaging incidence utilization by time since migration, stratified by age at migration.**
(PDF)

**S4 Table. Sex-specific relative rate of medical imaging utilization stratified by age at migration.**
(PDF)

**S5 Table. Relative rate of medical imaging utilization stratified by age at migration and migration year.**
(PDF)

**S6 Table. Relative rate of medical imaging utilization stratified by age at migration using an index date 2 years later.**
(PDF)

**S1 Fig. Average comorbidity score during the first 10 years of follow-up for migrant and nonmigrants.**
(PDF)

**S2 Fig. Utilization of head computerized tomography within 7 days of an emergency department visit for a head trauma.**
(PDF)

## Acknowledgments

Parts of this material are based on data and/or information compiled and provided by the Canadian Institute for Health Information (CIHI), Ontario Health (OH), and the Ontario Ministry of Health. Parts or whole of this material are based on data and/or information compiled and provided by Immigration, Refugees and Citizenship Canada (IRCC) current to May 2019. This document used data adapted from the Statistics Canada Postal CodeOM Conversion File, which is based on data licensed from Canada Post Corporation, and/or data adapted from the Ontario Ministry of Health Postal Code Conversion File, which contains data copied under license from Canada Post Corporation and Statistics Canada.

The analyses, conclusions, opinions and statements expressed herein are solely those of the authors and do not reflect those of the funding or data sources; no endorsement is intended or should be inferred.

The analyses, conclusions, opinions, and statements expressed in the material are those of the author(s) and not necessarily those of IRCC.

## Author Contributions

**Conceptualization:** Giancarlo Di Giuseppe, Jason D. Pole.

**Data curation:** Priscila Pequeno.

**Formal analysis:** Giancarlo Di Giuseppe, Priscila Pequeno.

**Funding acquisition:** Marilyn L. Kwan, Diana L. Miglioretti, Rebecca Smith-Bindman.

**Investigation:** Giancarlo Di Giuseppe, Marilyn L. Kwan, Diana L. Miglioretti, Rebecca Smith-Bindman, Jason D. Pole.

**Methodology:** Giancarlo Di Giuseppe, Rinku Sutradhar, Marilyn L. Kwan, Diana L. Miglioretti, Rebecca Smith-Bindman, Jason D. Pole.

**Visualization:** Giancarlo Di Giuseppe.

**Writing – original draft:** Giancarlo Di Giuseppe.

**Writing – review & editing:** Giancarlo Di Giuseppe, Rinku Sutradhar, Priscila Pequeno, Marilyn L. Kwan, Diana L. Miglioretti, Rebecca Smith-Bindman, Jason D. Pole.

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
