## [Editor Report · Decision Letter 0]

12 Feb 2024

Dear Dr Pole, 

Thank you for submitting your manuscript entitled "Immigrants medical imaging utilization compared to residents in the universal health-care system of Ontario, Canada" for consideration by PLOS Medicine.

Your manuscript has now been evaluated by the PLOS Medicine editorial staff and I am writing to let you know that we would like to send your submission out for external peer review.

However, before we can send your manuscript to reviewers, we kindly request you to:

1) Please ensure that the study is reported according to the STROBE guideline, and include the completed STROBE checklist as Supporting Information. Please add the following statement, or similar, to the Methods: ""This study is reported as per the Strengthening the Reporting of Observational Studies in Epidemiology (STROBE) guideline (S1 Checklist).""

2) we did note that the introduction and discussions section are very Canada focused. To appeal to our broad readership, please consider expanding these section and place your study and its results within the broader geographical context. 

We also need you to complete your submission by providing the metadata that is required for full assessment. To this end, please login to Editorial Manager where you will find the paper in the 'Submissions Needing Revisions' folder on your homepage. Please click 'Revise Submission' from the Action Links and complete all additional questions in the submission questionnaire.

Please re-submit your manuscript by the 15th of February.

Feel free to email me directly at kjanin@plos.org if you have any queries relating to your submission or like more time to re-submit your study to us.

Kind regards,

Katrien G. Janin, PhD

Senior Editor

PLOS Medicine

---

## [Decision Letter · Decision Letter 1]

18 Apr 2024

Dear Dr. Pole,

Thank you very much for submitting your manuscript "Immigrants medical imaging utilization compared to residents in a universal healthcare system" (PMEDICINE-D-24-00454R1) for consideration at PLOS Medicine. 

As you will see, the reviewers were positive about the paper but, they raised a number of questions about specific study details and the methodological approach. After discussing the paper with the editorial team and an academic editor with relevant expertise, I’m pleased to invite you to revise the paper in response to the reviewers’ comments. We plan to send the revised paper to some of all of the original reviewers*, and of course we cannot provide any guarantees at this stage regarding publication. 

When you upload your revision, please include a point-by-point response that addresses all of the reviewer and editorial points, indicating the changes made in the manuscript and either an excerpt of the revised text or the location (eg: page and line number) where each change can be found. Please submit a clean version of the paper as the main article file and a version with changes marked should as a marked-up manuscript. Please also check the guidelines for revised papers at http://journals.plos.org/plosmedicine/s/revising-your-manuscript for any that apply to your paper. 

We ask that you submit your revision by the 9th of May, 2024. However, if this deadline is not feasible, please contact me by email, and we can discuss a suitable alternative. 

Please don’t hesitate to contact me directly with any questions (kjanin@plos.org). If you reply directly to this message, please be sure to ‘Reply All’ so your message comes directly to my inbox. 

Kind regards, 

Katrien 

Katrien G. Janin, PhD 

PLOS Medicine 

plosmedicine.org

Kjanin@plos.org

*Please note: If your article is accepted, you may have the opportunity to make the peer review history publicly available. The record will include editor decision letters (with reviews) and your responses to reviewer comments. If eligible, we will contact you to opt in or out. 

Editorial comments: 

From an editorial perspective, we feel that the study is a bit superficial at the moment and that we feel there is potential for a much richer analysis, along the lines as suggestions by Reviewer 3. The Academic Editor also mentioned the following possibility "stratifying by having a PCP vs. not (there was a HUGE difference in this), index time (have things improved over secular time?), income quantile, geography, diagnosis codes, etc. rather than adjusting for all these, as stratifying and identifying interaction could have been somewhat more illuminating and may have spoken to underlying drivers a bit more." It is not necessary that you have to go about in that manner, but it is an option for helping to deepen the manuscript which we do feel is required.

In the Introduction, please add information on how the healthcare system works in Ontario, and the nirn al routed that people take to request/ be referred to receiving medical imaging. 

Please include your findings in a broader perspective (beyond the Ontarian/ Canadian Healthcare system) in your Discussion section. 

1. Data Availability: 

PLOS Medicine requires that the de-identified data underlying the specific results in a published article be made available, without restrictions on access, in a public repository or as Supporting Information at the time of article publication, provided it is legal and ethical to do so. Please see the policy at http://journals.plos.org/plosmedicine/s/data-availability and FAQs at http://journals.plos.org/plosmedicine/s/data-availability#loc-faqs-for-data-policy 

PLOS defines the “minimal data set” to consist of the data set used to reach the conclusions drawn in the manuscript with related metadata and methods, and any additional data required to replicate the reported study findings in their entirety. Authors do not need to submit their entire data set, or the raw data collected during an investigation. 

For each data source used in your study: 

2. Reporting guidance 

Please report your study according to the relevant guidance which can be found here https://www.equator-network.org/reporting-guidelines/

as your study is an observational study, please ensure that the study is reported according to the STROBE guideline, and include the completed STROBE checklist as Supporting Information. 

Please add the following statement, or similar, to the Methods: ""This study is reported as per the Strengthening the Reporting of Observational Studies in Epidemiology (STROBE) guideline (S1 Checklist).

When completing the checklist, please use section and paragraph numbers, rather than page numbers."

For all observational studies, in the manuscript text, please indicate: (1) the specific hypotheses you intended to test, (2) the analytical methods by which you planned to test them, (3) the analyses you actually performed, and (4) when reported analyses differ from those that were planned, transparent explanations for differences that affect the reliability of the study's results. If a reported analysis was performed based on an interesting but unanticipated pattern in the data, please be clear that the analysis was data-driven.

3. Statistical reporting 

As per standard PLOS Medical requirements, please provide 95% CIs and p values for all results where appropriate (including the abstract). We suggest reporting statistical information in the following format: ‘x’; (95% CI [‘y’,’ z’] p value) and use commas as opposed to hyphens (as these can be confused with negative values) to separate upper and lower bounds. For p values, please report as p<0.001 and where higher as 'p=0.002'. Please add the statistical method used to your method section. We also invite you to report p values to consistently to the third decimal digit - thousandths.

If applicable, please include any important dependent variables that are adjusted for in the analyses.

4. Title 

Please revise your title according to PLOS Medicine's style. Your title must be nondeclarative and not a question. It should begin with main concept if possible. "Effect of" should be used only if causality can be inferred, i.e., for an RCT. Please place the study design ("A randomized controlled trial," "A retrospective study," "A modelling study," etc.) in the subtitle (ie, after a colon). 

4. Abstract layout 

Please structure your abstract using the PLOS Medicine headings (Background, Methods and Findings, Conclusions). 

5. Author summary 

At this stage, we ask that you include a short, non-technical Author Summary of your research to make findings accessible to a wide audience that includes both scientists and non-scientists. The authors summary should consist of 2-3 succinct bullet points under each of the following headings: 

• Why Was This Study Done? Authors should reflect on what was known about the topic before the research was published and why the research was needed. 

• What Did the Researchers Do and Find? Authors should briefly describe the study design that was used and the study’s major findings. Do include the headline numbers from the study, such as the sample size and key findings. 

• What Do These Findings Mean? Authors should reflect on the new knowledge generated by the research and the implications for practice, research, policy, or public health. Authors should also consider how the interpretation of the study’s findings may be affected by the study limitations. In the final bullet point of ‘What Do These Findings Mean?’, please describe the main limitations of the study in non-technical language. 

The Author Summary should immediately follow the Abstract in your revised manuscript. This text is subject to editorial change and should be distinct from the scientific abstract. Please see our author guidelines for more information: https://journals.plos.org/plosmedicine/s/revising-your-manuscript#loc-author-summary

6. Introduction layout 

Please address past research and explain the need for and potential importance of your study. Indicate whether your study is novel and how you determined that. If there has been a systematic review of the evidence related to your study (or you have conducted one), please refer to and reference that review and indicate whether it supports the need for your study. 

7. Discussion layout 

Please present and organize the Discussion as follows: a short, clear summary of the article's findings; what the study adds to existing research and where and why the results may differ from previous research; strengths and limitations of the study; implications and next steps for research, clinical practice, and/or public policy; one-paragraph conclusion. 

8. Supplementary materials 

Please note that supplementary materials are not checked and will be posted as supplied by the authors. Therefore, please double check. Please cite your Supporting Information as outlined here: https://journals.plos.org/plosmedicine/s/supporting-information - Please note you may use almost any description as the item name of your supporting information as long as it contains an "S" and number. For example, “S1 Appendix” and “S2 Appendix,” “S1 Table” and “S2 Table. Please ensure each supplementary material has a call out (link) from your main manuscript. 

Comments from the reviewers:

Reviewer #1: Thanks for the opportunity to read your manuscript. My role is statistical reviewer, so I have focused on the design, data, and analysis that are presented. I have put general comments first, followed by questions relevant to a specific section of the manuscript (with a page/line reference). 

This manuscript examines differences between immigrants and non-immigrants in utilisation of medical imaging. The study uses routinely collected health data from Ontario, Canada and includes all residents registered for health services in this province between 1995 and 2016. This data was linked to a national immigration database. The main analyses used a matched sample of immigrants to non-immigrants, using a pool of potential matches created per year. Matching was based on age at time, sex, and area. Main models comparing utilisation between immigrants and non-immigrants included a measure of comorbidity burden, age, sex, and use of primary care. The main analysis used recurrent event survival analysis - this is appropriate I agree with the authors that this has advantages over using a generalised linear model with a count distribution. Participants with missing covariate information were retained in the analyses using the 'missing as indicator' approach which assumes that this data is missing-completely-at-random. Several sensitivity analyses are considered, including starting follow-up 2 years later so the same approach to measuring comorbidity is used in both immigrants and non-immigrants, and the use of head CT in participants experiencing a TBI, which is not expected to show a difference between immigrants and non-immigrants.

Using a different temporal window to allow for a lookback period to determine comorbidities in immigrants does not make an appreciable difference to the main results. This is an interesting study - results are well presented, and I think the analysis is good (a few queries below).

P8, L159. So for immigrants, comorbidity status is concurrent with imaging utilisation, while for non-immigrants this precedes measurement of utilisation. Given that healthcare utilisation is driven by need, does this inappropriately adjust for current health status in immigrants and drive estimated differences between immigrants and non-immigrants? 

P8, L163. What information did you use to estimate that comorbidity status would not change within the first 2 years of arrival? 

P6, L117. Were non-immigrants taken out of the pool of possible matches once matched? 

P6, L118. PLOS medicine has an international readership, what size (area and pop) would a typical forward sortation area have? 

 P6, L140. Are there any private imaging services in this province that might not be captured by the OHIP? 

P8, L177. Does 'accounted' mean that this was achieved as a result of prior matching, or because there was stratification or adjustment when estimating the rate differences?

P8, L186. Were participants censored at death? Was there any way to determine when participants moved from Ontario and no longer at risk? 

P8, L191. How much covariate information was missing from the data? Were those missing covariate data similar to those with complete data? i.e. is it reasonable to assume that this data is missing-completely-at-random? 

P10, L213. Was this because they left the region? 

P13, L272. I think another word other than 'necessary' could be used here, I am sure many of the imaging services provided other than head CTs were necessary to the participants. 

L13, L272. If immigrants presenting to the ED with head trauma are more likely to receive a head CT (26% vs 18%), but the overall number of CTs per trauma is similar between immigrants and non-immigrants, does this mean the distribution of the number of head CTS is different (i.e. non-immigrants more likely to get multiple head CTs)? 

Reviewer #2: Thank you, editor, for sending this paper to read.

This paper raised good research question, well designed methodology and analysis and interpretation. Such findings are expected the people from immigrant's background have to face several countries in the destination countries, and always low service utilization compared to local population.

The utilization of medical imaging is depending on the diseases prevalence. Without knowing patterns of disease, we cannot say that there is disparity of medical imaging services utilization. 

As authors mentioned migrants have more healthy migrant effects. Authors should mention in the limitation of the study or provide data for this.

I hope this feedback help to revise this work.

Reviewer #3: Reviewer's Report - 

Immigrants medical imaging utilization compared to residents in a universal healthcare system

Review for PLOS Medicine

Manuscript number: PMEDICINE-D-24-00454R1

Summary: This paper presents an empirical investigation as to how the usage of medical imaging healthcare by 'immigrants' in the state of Ontario, Canada, compares to residents. The work benefits from a rich administrative data set - with 21 years of data 1995-2016 of complete records for CT, MRI, radiography, and ultrasonography utilisation of patients - with important patient information including information on gender, age, comorbidity which allows for controlling of key explanatory variables. The statistical methodology is appropriate and well-executed - ways of backfilling data etc to deal with data gaps are appropriate, the sensitivity analysis with CT utilisation for traumatic brain injury is very nice. The Tables and charts are well presented. I have some concerns about how some aspects of the paper are described - and have additional suggestions to improve the contextual understanding and presentation of the work for an international readership - outlined in this peer- review. 

Overall, I see the merit in the research, and recommend revisions of the content and presentation of the work undertaken which where address can allow for the paper to progress towards publication in PLOS Medicine. 

Title/Abstract - Throughout this paper (and even the labelling on your charts!) I do not agree with the distinction between the groups as immigrants versus residents of Canada. The immigrant groups do in fact appear to be residents - so surely it is more accurate to describe the groups as native versus non-native residents of Canada - from what I gather they are distinguished by country of birth i.e. nationality. 

The reference to matching 1 to 1 on residents based on age, sex, immigration year (?), and geography is confusing - how can you match native residents versus immigrant residents based on immigration year - this does not make sense to me at least on first read? Please clarify if this is a typo or something else?

The results accord with expectations apart from the result on the longer the time since immigration the larger the gap - interesting and slightly counterintuitive.

Keywords- I don't think the use of emigrants under this paper is appropriate - also I do not think this is a traditional cohort study - so this terminology is misleading (?)

Introduction - please provide clarification on your resident and migrant distinction. 

Since Canada is a universal healthcare system in which GPs are the main gatekeepers - and GPs can be a key route through which patients access medical imaging, it is important to also cite literature which examines immigrants access to GP services - e.g. Barlow, Mohan, and Nolan (2022) Utilisation of healthcare by immigrant adults relative to the host population: Evidence from Ireland. Journal of Migration and Health. 

Line 85: radiology use or access

Line 87: crucial to gain insight into any disparities in immigrant populations for policy making and healthcare planning… Please refer to important policy documents from the WHO in this area e.g. World Health Organization, 2019. Promoting the Health of Refugees and Migrants Draft Global Action Plan, 2019-2023. https://apps.who.int/gb/ebwha/pdf_files/WHA72/A72_25-en.pdf. 

It would also be good to know if the Canadian government or a Canadian Medical Association - or locally in Ontario had policies published on migrant health - there is no enough discussion of policies and planning in this area - particularly if there is any consideration of this in terms of radiology which is a unique selling point of your paper - you do not sufficiently sell the special case of native v non-natives use of medical imaging as a motivation for your research - the use or non-use of this aspect of healthcare obviously has downstream effects e.g. lack of detection/diagnosis leads to poorer patient outcomes (more advanced cancers etc), preventable mortality etc . - provide references to strengthen your arguments. Greater sell the novelty and wider importance of this work.

The work also does not provide much consideration of the world beyond the Canadian context - how generalisable are the findings? What lessons can be gleemed for other countries/regions and jurisdictions?

You do not provide a clear research question - no outline a clear aim of the research. The hypothesis you provide is not supported by any other empirical references or a previous theoretical underpinning - this is needed to strengthen your argument. - Is it an assimilation or integration argument that you are making - the longer a person is resident in a country they adopt the norms of the local population, including healthcare utilisation norms?

And then in testing this hypothesis I see later you have categories of >10 years in Canada, > 20 years etc - is the hypothesis based on any particular cut off period (e.g. over 10/20 years) or just a gradient over time should be seen?

Methods

You have really comprehensive coverage on migrants. I am uncomfortable with the use of the term 'landed' throughout - maybe arrival/taking up residency is a better way of phrasing this?

It is not clear whether or not you have the exact country or region of origin of migrants in your dataset - if not, please clarify (maybe I have missed this) - the literature has shown that there can be significantly heterogenous findings on healthcare utilisation by different migrant groups - and you are treating them all as a homogenous group. For example, Mohan (2021) The influence of caregiver's migration status on child's use of healthcare services: evidence from Ireland, Sociology of Health and Illness, finds different patterns in use of healthcare services by children of different migrant groups e.g. those from less advanced non-English speaking non-European countries availed less of GP services less than native population - and other types of migrant groups. This paper may also be relevant for commentary on your 0-19 year old group - since children (of migrant background) rely on their parents/caregivers to help them access (and are involved in the decision making process around) healthcare services - and the parents are likely to be of a migrant background themselves. 

You may need to better describe in the Introduction the typical pathways for a patient to access medical imaging services. For example, describing the GP as gatekeeper role - which I assume is free at the point of use (or is there any co-payments) and then if there are co-payments in services - the reason I query this here is you say that a source of your medical imaging data is from fee-for-service physician billing records (OHIP Schedule of Benefits and Fees) - and an international readership will not understand what this is under a universal healthcare system. In the UK's NHS which is a universal healthcare system there are no fees for services - e.g. copayments - so outlining the institutional setting is important. 

Please define meaning of ICES.

Is your control variable of a visit to a primary care provider a binary variable - and is that in the prior 12-month period of what is the exact definition?

Discussion

Your surprising result which is nice that you can test in the data that you have (many studies do not have duration of residence in the country) - was that the longer a person is resident in Canda there medical imaging use remained lower than that of their Canadian counterparts - more could be commented on this again bringing in theory or supporting references on lack of assimilation/integration. The point you make that 'passively waiting for access for new immigrants to improve over time does not occur' is a key indication from this paper - to which more could be made of this in terms of evidence gathering to inform national government policies and the like of supranational organisations such as the WHO, UN, EU etc. 

The point on older migrant groups accessing services less is also insightful - you do not have data on whether they have children or not - but again the family circumstances of the household are likely to be key here (as in the case of children previously mentioned) - so while you can't answer this question it may be that older immigrant people with children may be better able to access services if there is a language barrier etc. - as the children can speak on their behalf - i.e. there may even be heterogenous access within this group depending on family circumstances. 

A reason for lower utilisation of services by immigrant groups which has not been mentioned is that sometime immigrant groups go back to their home country of origin of care e.g. Wallace et al (2009) showed that Mexican migrants in the US went home to Mexico to receive care - this may not be to the same extent in the Canadian setting. 

Again, the policy and practice implications and the relevance for an international audience needed greater consideration.

 Overall, great data and well executed analysis which could be valuable where the paper is strengthened. 

Reviewer #4: I wish to congratulate the authors in undertaking this ambitious study that illustrates diverging patterns of imaging completion among immigrants compared to non-immigrant groups. The data and modeling are well done, but its ability to be interpreted depends on how it's contextualized. As such I feel that the paper could benefit from better framing and slightly more analysis. 

First, it's a bit unclear why, given the author's well-placed literature about disparities in radiography and its multifaceted drivers why the authors' hypothesis is that in Canada disparities would decrease with time in Canada. Do they presume that in Canada the limiting factor is unmeasured system navigational skills? 

Related to the above point, it is important to emphasize what the authors have identified is imaging *completion*, which as they hint in the discussion, is essentially a function of differences in health care access, clinic/inpatient utilization, provider driven ordering, patient acceptance, and patient follow-through. The universal health system in Ontario provides good access to care and while there are very wide post-match disparities in clinic visits as seen in Table 1, ideally that is well controlled for in models. I assume the authors don't know about uncompleted orders, although I interpret the fact that overall imaging rates only diverge after more time in Canada as arguing against systematic cultural resistance to imaging. 

However, there is little interrogation about whether providers are choosing to offer radiological tests to older immigrants in lower amounts in outpatient settings (presuming that outpatient ordering accounts for a substantial amount of imaging in this sample), particularly as there is substantial literature on provider-driven discrimination in ordering. 

I think that this paper would be strengthened by 1) a conceptual model (which I strongly recommend) and 2) a sensitivity analysis to investigate discrimination in provider ordering. As for models, there are several out there, but any model should at the very least undergird the importance of the question and support your hypothesis that there is convergence in imaging rates with time. Having a model (or at least explicitly delineating challenges along the patient need to imaging completion pipeline) might also help frame paragraph 2 and 3 of discussion explaining this gap. 

As for the second item, I understand that the team does not have detailed indication data, but it seems they do have ICD or similar diagnoses. It would be useful to complement the Emergency Room CT Scan analysis with a sensitivity analysis that identifies imaging rates for a diagnosis attached to a common incident complaint (e.g., CT/MRI head for incident headache over 65, a "red flag"). This example isn't prescriptive, of course, but simply an example of what the authors may attempt. 

Finally, it would be useful, even as a supplementary table, to include an interaction with sex to your Table 2 Model, to identify intersectional gender-immigration disparities, which are probably quite important. 

Several minor issues: 

Page 7, line 146: The sentence "Utilization for CT, MRI, radiography, and ultrasonography modalities were evaluated daily for all examination types". Some clarification is needed by "evaluated daily". If you're simply prefacing the deduplication efforts, you might consider removing the word "daily" above to avoid confusion. 

Page 13, line 267: "Older adult immigrants had slightly higher RR of ultrasonography" (RR, 0.98; 95%CI: 0.98, 0.98). The RR is less than 1—it should be "lower". 

Page 15, line 312-314: I presume "increased" should be "increase". Please also refer to Table 2 again here. Also, this hypothesis should be easily proven by disaggregating by sex, if not ultrasound type?

General journal requests: 

1. Please upload any figures associated with your paper as individual TIF or EPS files with 300dpi resolution at resubmission; please read our figure guidelines for more information on our requirements: http://journals.plos.org/plosmedicine/s/figures. While revising your submission, please upload your figure files to the PACE digital diagnostic tool, https://pacev2.apexcovantage.com/. PACE helps ensure that figures meet PLOS requirements. To use PACE, you must first register as a user. Then, login and navigate to the UPLOAD tab, where you will find detailed instructions on how to use the tool. If you encounter any issues or have any questions when using PACE, please email us at PLOSMedicine@plos.org. 

To submit your revised manuscript please use the following link: 

[LINK]

---

## [Decision Letter · Decision Letter 2]

18 Jul 2024

Dear Dr Pole,

Many thanks for submitting your manuscript "Medical imaging utilization in immigrants compared with non-immigrants in a universal healthcare system: a population-based matched cohort study" (PMEDICINE-D-24-00454R2) to PLOS Medicine. The paper has been reviewed by subject experts and a statistician; their comments are included below and can also be accessed here: [LINK]

Thank you for your detailed response to the editors' and reviewers' comments. I have discussed the paper with my colleagues and the academic editor, and it has also been seen again by the original reviewers. The changes made to the paper were satisfactory to the reviewers. However, the editorial team concurs with the Academic Editor to request a some additional details, as listed below. Therefore, we ask you to carefully address the comments in a further revision to preclude the need for further revisions. When submitting your revised paper, please again include a detailed point-by-point response to the comments.s

We ask that you submit your revision by Aug 01 2024 11:59PM. However, if this deadline is not feasible, please contact me by email, and we can discuss a suitable alternative.

Don't hesitate to contact me directly with any questions (kjanin@plos.org). 

Best regards, 

Katrien 

Katrien Janin, PhD 

Associate Editor

PLOS Medicine

kjanin@plos.org

Comments from the Academic Editor and editorial Team

Thank you very much for your revised manuscript. We all agree that the manuscript is much strengthened.

1. The academic editor and the editorial team are interested in seeing interactions across having PCP or not and income quartile. We don't think simple adjustment in a model is sufficient to address how these complex variables interplay with access to healthcare.

2. The academic editor also commented that they would still be interesting for you to frame your results with something like the Levesque framework for access to healthcare.

3. We also like to suggest you use the word “migrant” rather than “immigrant”. 

4. Data Availability Statement: please revise your statement, as authors cannot be the contact person. I also like to encourage you to make the code openly available, you may wish to deposit this in Github or another repository. If code cannot be shared due to legal or ethical reasons then authors should state this in the Data Availability Statement and give details of how to request access to the code (please note that once again, authors can not be the contact person). You can find more information here: https://journals.plos.org/plosmedicine/s/materials-software-and-code-sharing

5. Figures: I noticed you use red and green in your figures. Please consider avoiding the use of red and green in order to make your figure more accessible to those with colour blindness. 

6. PLOS Style: Discussion layout. Please present and organize the Discussion as follows: a short, clear summary of the article's findings; what the study adds to existing research and where and why the results may differ from previous research; strengths and limitations of the study; implications and next steps for research, clinical practice, and/or public policy; one-paragraph conclusion. Please remove all other sub headers. 

7. Supplementary materials. This is a kind reminder that supplementary materials are not checked and will be posted as supplied by the authors. Therefore, please double check. Please cite your Supporting Information as outlined here: https://journals.plos.org/plosmedicine/s/supporting-information - Please note you may use almost any description as the item name of your supporting information as long as it contains an "S" and number. For example, “S1 Appendix” and “S2 Appendix,” “S1 Table” and “S2 Table. Please ensure each supplementary material has a call out (link) from your main manuscript.

Comments from the Editorial Team:

Comments from the reviewers: 

Reviewer #1: Thanks for the revised manuscript and responses to my original review. The updated manuscript resolves my original queries, this is an interesting study that I enjoyed reviewing. 

I agree with the authors that excluding missing data when the amount is ~0.2% is reasonable for socioeconomics and immigrant category. 

Reviewer #3: The authors have adequately addressed the comments of my previous review - based on my own assessment of the paper, I recommend the revised version of the manuscript for publication 

Reviewer #4: I appreciate the authors taking the time to revise their report according to recommendations. A few further items to consider:

1) Page 9, line 157: I would not consider matching to be random process--the probability is being chosen is uniform only given observed characteristics, not unobserved ones. I would remove the word "randomly'.

2) The Anderson-Gill model is a type of survival analysis, and estimates should be reported as hazard ratios. (https://academic.oup.com/ije/article/44/1/324/654595).

3) Lines 366-369: Rather than simply say that it is unlikely that immigrants are getting imaging outside of Canada, I would consider offering up the fact that you are matching on 2-year primary care utilization; if they are seeing the doctor in similar amounts in country they should be getting imaging in country in similar amounts as well, theoretically. (Getting additional healthcare/imaging while traveling/living abroad--as many elderly immigrants with regularized immigration status do--is still possible, but less likely).

---

## [Editor Report · Decision Letter 3]

27 Aug 2024

Dear Dr. Pole,

Thank you very much for re-submitting your manuscript "Medical imaging utilization in migrants compared with non-migrants in a universal healthcare system: a population-based matched cohort study" (PMEDICINE-D-24-00454R3) for review by PLOS Medicine.

I have discussed the paper with my colleagues and the academic editor and it was also seen again by the reviewers. I am pleased to say that provided the remaining (minor) editorial and production issues are dealt with we are planning to accept the paper for publication in the journal.

[LINK]

Please don’t hesitate to contact me directly with any questions (kjanin@plos.org). If you reply directly to this message, please be sure to ‘Reply All’ so your message comes directly to my inbox. 

We look forward to receiving the revised manuscript by Sep 03 2024 11:59PM.   

Sincerely,

Katrien Janin, PhD

Senior Editor 

PLOS Medicine

plosmedicine.org

Requests from the Academic Editor and Editors:

Thank you very much for your revised manuscript. We find it much improved and only have minor requests at this stage:

1) The Academic Editor is still not entirely convinced by your response to the interaction comment. We propose that you are explicit, and consider changing the sentence to:

"This suggests a need for future research to explore other mechanisms which may aid in explaining the lowered utilization of medical imaging or if there is heterogeneity in the effects of being a migrant on utilization across factors are effect modifiers such having a PCP or income.”

2) In the ‘Abstract Methods and Findings’ section, in the last sentence of the Abstract Methods and Findings section, please describe the main limitation(s) of the study's methodology.

Comments from Reviewers:

Reviewer 1:

Thanks for the revised manuscript and responses to my original review. The updated manuscript resolves my original queries, this is an interesting study that I enjoyed reviewing.

I agree with the authors that excluding missing data when the amount is ~0.2% is reasonable for socioeconomics and immigrant category.

Reviewer 3:

The authors have adequately addressed the comments of my previous review - based on my own assessment of the paper, I recommend the revised version of the manuscript for publication

Reviewer 4:

I appreciate the authors taking the time to revise their report according to recommendations. A few further items to consider:

1) Page 9, line 157: I would not consider matching to be random process--the probability is being chosen is uniform only given observed characteristics, not unobserved ones. I would remove the word "randomly'.

2) The Anderson-Gill model is a type of survival analysis, and estimates should be reported as hazard ratios. (https://academic.oup.com/ije/article/44/1/324/654595).

3) Lines 366-369: Rather than simply say that it is unlikely that immigrants are getting imaging outside of Canada, I would consider offering up the fact that you are matching on 2-year primary care utilization; if they are seeing the doctor in similar amounts in country they should be getting imaging in country in similar amounts as well, theoretically. (Getting additional healthcare/imaging while traveling/living abroad--as many elderly immigrants with regularized immigration status do--is still possible, but less likely).

[LINK]

---

## [Editor Report · Decision Letter 4]

15 Sep 2024

Dear Dr Pole, 

On behalf of my colleagues and the Academic Editor, Aaloke Mody, I am pleased to inform you that we have agreed to publish your manuscript "Medical imaging utilization in migrants compared with non-migrants in a universal healthcare system: a population-based matched cohort study" (PMEDICINE-D-24-00454R4) in PLOS Medicine.

PRESS

Thank you again for submitting to PLOS Medicine, and we look forward to publishing your paper. Should any questions arise during the post-accept process, please feel free to reach out to me directly (hvanepps@plos.org), as Katrien Janin has now left the journal. 

Kind regards,

Heather

Heather Van Epps, PhD

Executive Editor, PLOS Medicine

[on behalf of]

Katrien G. Janin, PhD 

Senior Editor 

PLOS Medicine